# The Intersection of a Child’s Demographics and Household Socioeconomic Status in the Multimorbidity of Malaria, Anaemia, and Malnutrition among Children Aged 6–59 Months in Nigeria

**DOI:** 10.3390/ijerph21050645

**Published:** 2024-05-19

**Authors:** Phillips Edomwonyi Obasohan, Stephen J. Walters, Richard M. Jacques, Khaled Khatab

**Affiliations:** 1School of Medicine and Population Health, University of Sheffield, Sheffield S1 4AD, UK; s.j.walters@sheffield.ac.uk (S.J.W.); r.jacques@sheffield.ac.uk (R.M.J.); 2Department of Liberal Studies, College of Administrative and Business Studies, Niger State Polytechnic, Bida Campus, Bida 912231, Nigeria; 3Faculty of Health and Wellbeing, Sheffield Hallam University, Sheffield S10 2BP, UK; k.khatab@shu.ac.uk

**Keywords:** multiple diseases, moderation effects, predictive margins, under five, interactions, syndemic

## Abstract

Multimorbidity of malaria, anemia, and malnutrition (MAMM) is a condition in which an individual has two or more of these health conditions, and is becoming an emergent public health concern in sub-Saharan African countries. The independent associations of a child’s demographic variables and household socioeconomic (HSE) disparities with a child’s health outcomes have been established in the literature. However, the effects of the intersection of these factors on MAMM, while accounting for other covariates, have not been studied. Therefore, this study aimed to determine how children’s sex, age, and household socioeconomic status interact to explain the variations in MAMM among children aged 6–59 months in Nigeria. Data from the 2018 Nigeria Demographic and Health Survey and the 2018 National Human Development Report (NHDR) were used. This study included weighted samples of 10,184 children aged 6–59 months in Nigeria. A three-level multilevel mixed effect ordinal logistic regression model was used, such that individual characteristics at level 1 were nested in communities at level 2 and nested in states at level 3. Subsequently, predictive probability charts and average adjusted probability tables were used to interpret the intersectional effects. Five models were created in this scenario. Model 1 is the interaction between the child’s sex and household wealth status; model 2 is the interaction between the child’s sex and age; model 3 is the interaction between the child’s age and household wealth status; model 4 has the three two-way interactions of the child’s sex, age, and household wealth status; and model 5 includes model 4 and the three-way interactions between a child’s sex, age, and household wealth quintiles; while accounting for other covariates in each of the models. The prevalence of children with a ‘none of the three diseases’ outcome was 17.3% (1767/10,184), while 34.4% (3499/10,184) had ‘only one of the diseases’, and 48.3% (4918/10,184) had ‘two or more’ MAMMs. However, in the multivariate analyses, model 3 was the best fit compared with other models, so the two-way interaction effects of a child’s age and household wealth status are significant predictors in the model. Children aged 36–47 months living in the poorest households had a probability of 0.11, 0.18, and 0.32 of existing with MAMM above the probability of children of the same age who live in the middle class, more prosperous, and richest households, respectively, while all other covariates were held constant. Thus, the variation in the prevalence of MAMM in children of different ages differs depending on the household wealth quintile. In other words, in older children, the variations in MAMM become more evident between the richer and the poorer household quintiles. Therefore, it is recommended that policies that are geared toward economic redistribution will help bridge the disparities observed in the prevalence of multiple diseases among children aged 6–59 months in Nigeria.

## 1. Introduction

Multimorbidity of anemia, malaria, and malnutrition (MAMM) is a condition in which an individual has two or more of these health outcomes, and is becoming an emergent public health concern in sub-Saharan African countries. A recent study conducted in Nigeria found that one in every two children aged 6–59 months has two or more MAMMs [1]. Similarly, other findings showed that 68.1%, 35.5%, and 43.6% of children aged 6–59 months in Nigeria are anemic [2], malaria positive [3], and malnourished [4], respectively. In low and middle-income countries (LMICs), results from past studies have often pointed out that gender variation in childhood health outcomes exists strongly at the advantage of the girl-child [5,6,7,8,9,10,11,12,13]. The reason for this has not been explicitly concluded. Also, older children are found to be more prone to diseases when compared to their younger siblings [14,15,16,17,18,19,20,21], just as it is often concluded that children from poor households are more at risk of childhood diseases than those from wealthy households [21,22,23,24,25]. In recent times, social stratifications along wealth status have become even more evident in many developing countries, including Nigeria. In terms of diseases among Nigerian children, variations in social classifications have often been between the rich and the poor. So, it will be interesting to know whether gender-sensitive disparities in the age stratum are a result of differences in household socio-economic status (HSES). Therefore, the fundamental tenet of intersectionality theory is the conceptualization of these various social identities [26]. The presence of an interaction effect occurs when a second independent variable moderates the relationship between the first independent variable and the dependent variable [26,27]. For instance, when there is a variation in the relationship between a child’s sex and poor health and a child’s age or household socioeconomic status (HSES), it means there is an interaction effect between the child’s sex and age or HSES and the child’s adverse health outcomes. The implications of this are that the ways a boy-child and a girl-child may experience MAMM may differ for different HSESs.

Meanwhile, in Nigeria, the independent associations of a child’s sex, age, and household socioeconomic disparities with a child’s independent health and multiple health outcomes have been established in the literature [1,28]. However, the effects of the intersection of these factors have not been reported in the literature. In order to determine how gender, age, and socioeconomic status interact to explain disparities in MAMM among children aged 6–59 months in Nigeria, the intersectionality theory was used in this study. Moreover, when there are several forms of inequality within a social group, policies meant to reduce health disparities between them may be ineffective [26]. Therefore, the aim of this study is to investigate how a child’s sex, age, and household socioeconomic status interact in the presence of other covariates to explain the variations in the prevalence of MAMM among children aged 6–59 months in Nigeria.

## 2. Methods and Data

### 2.1. Sampling Techniques and Sample Size

This is a secondary analysis of two cross-sectional survey data sets, the 2018 Nigeria Demographic and Health Survey (NDHS), and the 2018 National Human Development Report (NHDR). The NDHS served as the main data, while contextual (state level) covariates were extracted from the NHDR and merged into the NDHS using a state unique identifier. Two stages of stratified cluster sampling were employed by the NDHS. First, a proportionate allocation stratified by place of residence (rural and urban) was used to pick a sample of enumeration areas (EAs) independently from each stratum. At the second stage, households were picked from the designated EAs using a systematic sampling technique [29]. This study included weighted samples of 10,184 children aged 6–59 months in Nigeria.

### 2.2. Variables Definitions

In the NDHS, data on the prevalence of anemia, malaria, and malnutrition were used in accordance with the World Health Organization standards. Anemia data were kept in the kid recode file (KR), and malaria and malnutrition data were stored in the household members recode file (PR). The PR file was then merged into KR files. However, the classification of each of these health outcomes has been reported in previous studies [2,3,4]. Therefore, to allow for multi-categorical responses represented in three intersecting sets of anemia, malaria, and malnutrition, the multimorbidity (with three outcomes) was grouped into eight separate categories (none of the diseases; anemia only; malaria only; malnutrition only; anemia and malaria; anemia and malnutrition; malaria and malnutrition; and anemia, malaria, and malnutrition), which were further reclassified to conform with the definition of multimorbidity counts of 0, 1, and 2, representing ‘no disease’, ‘one disease only, and ‘two or more diseases’, respectively.

### 2.3. Independent Variables and Rationale

The main independent variables of interest were the two-way and three-way interactions of a child’s age (categorized into 5 ordinal groups: 6–11, 12–23, 24–35, 36–47, and 47–59 months), sex (categorized as male or female), and household wealth quintiles (categorized into poorest, poorer, middle class, richer, and richest). The rationale for the choice of these variables was on the premises of findings in previous studies to determine the predictors of each of the outcome variables in MAMM (i.e., anemia, malaria, and malnutrition), among children aged 6–59 months in Nigeria. These variables (child’s sex, age, and household socioeconomic status), were outstandingly common and statistically significant factors found in the literature [2,3,4]. In addition, considering the enormous current social and economic challenges in Nigeria, resulting in social classifications being redefined [30], the position of the interactions with a child’s demographics on adverse health outcomes becomes necessary. The covariates accounted for in this study included child-related (sex, age, birth size, preceding birth interval, the child took iron supplements, duration of breast feeding, the child was dewormed, the child had a fever, and the child’s place of delivery); parental-related (maternal highest education, the mother resides with a partner, the mother’s religious status, the mother’s anemia status, paternal work status, and paternal education status); household-related (household wealth status, the under-five-year-old child slept under a bed net, sex of the household head, and the number of persons in the household); community-related (community wealth level, level of proportion of those whose community distance to a health facility is no big problem, and proportion of community maternal education level); and state/area-related (state multidimensional poverty index, state human development index, state gender inequality index, state region of residence, and place of residence) variables. Furthermore, the definitions, classifications and selections processes of these covariates have been reported elsewhere [3].

### 2.4. Model Specification

Five models were created in this scenario while accounting for the covariates considered in the study. Model 1 included the interaction between a child’s sex and household wealth status; model 2 was the interaction between a child’s sex and age; model 3 was the interaction between a child’s age and household wealth status; model 4 had the three two-way interactions of child’s sex, age, and household wealth status; and model 5 included model 4 and the three-way interactions between a child’s sex, age, and household wealth quintiles. In each of these models, the main effects of sex, age group, household wealth status, and all other covariates were accounted for. The model of best fit was determined using the model with the highest log-likelihood, and the lowest Akaike Information Criteria (AIC) and Bayesian Information Criteria (BIC).

### 2.5. Data Analysis

Three analytical procedures were used to answer the aims of this study.

In view of the hierarchical nature of the data sets, coupled with the ordinal classifications of the outcome variable, a three-level mixed effect ordinal logistic regression (MMEOLR) was used to determine the significance of the predictors, such that the individual child and household variables at level 1 were nested in the community at level 2 and were nested in the state at level 3. The three-level model was found to be more appropriate than the two-level model or a single-level model by the application of a likelihood ratio test (the likelihood ratio test for level-2 nested in level-3). This study involved fitting three-level mixed-effects ordinal logistic regression models, and testing the violation of the proportional assumption is difficult and not straightforward, especially when employing *brant* and *omodel* tests in Stata, following a mixed-effect ordered logistic regression. However, a ‘naïve’ approach (explained elsewhere [1]) was applied and it was found that there was no violation in the test. The results using the MMEOLR are displayed in Appendix A.To better understand the associations of the intersecting independent variables, interaction plots [27] were drawn for each intersection.Furthermore, because of the difficulties in the interpretations of intersectionality in terms of odd ratios which the MMEOLR presents, probabilities tables were generated to complement the interaction plots [31,32,33].

The computations were performed using STATA 17 (StataCorp LP: College Station, TX, USA), and a *p*-value less than 0.05 was considered significant for the predictors.

## 3. Results

Table 1 shows the distributions and associations between a child’s sex, age group, and household wealth status with the prevalence of MAMM among Nigerian children aged 6–59 months. The prevalence of children with ‘none of the three diseases’ outcomes was 17.3% (1767/10,184), while 34.4% (3499/10,184) had ‘only one of the diseases’, and 48.3% (4918/10,184) had ‘two or more’ MAMMs. Overall, 51.2% (5217/10,184) of the children in the multimorbidity sample were male. Consequently, the percentage of male children with two or more of the conditions was 50.62% (2641/5217), whereas the percentage for female children was 45.8% (2277/4967). The age group of 12 to 23 months had the highest percentage of children with two or more diseases (multimorbidity), at 51.4% (1245/2422), and the largest number of children overall, at 23.8% (2422/10184). Children in the 24–35-month age group had the next highest prevalence of two or more diseases at 51.1% (1102/2160).

Children from homes with the poorest household wealth index quintile made up the largest proportion of those with MAMM, with 73.6% (1393/1893) living with two or more diseases, followed by the poorer household wealth quintile at 63.8% (1268/1989). The proportion of children with two or more diseases appeared to decrease as the household wealth index quintile increased. The Chi-square test shows bivariately that the three principal predictors (sex, age group, and household wealth quintile) were strongly associated with the distribution of MAMM. The results of the multivariate analyses of the interactions and covariates in odds ratios using a three-level mixed effect ordinal logistic regression (MMEOLR) are contained in Appendix A.

### 3.1. Model Comparison and Fit

Table 2 shows the model comparison indicators. Model 3, which contained the interactions between a child’s age and household wealth status while accounting for all the covariates, was adjudged the best fit model using the least AIC (13,977.3) and the best combination of log-likelihoods and BICs.

### 3.2. Accounting for the Intersections

Figure 1a illustrates the predictive margins plot of the interaction between a child’s sex and household wealth quintiles on the prevalence of MAMM among children aged 6–59 months in Nigeria. The three bands (from bottom to top) represent the outcome status of ‘none of the diseases’, ‘one of the diseases’, and ‘two or more of the diseases’. Each line in the band represents the wealth quintiles. In the upper band, for instance, the probabilities of female children with MAMM were generally lower compared with their male counterparts, and the variations differed, but were not statistically significant across the household wealth status. So, the parallel lines indicate the non-presence of interaction effects between a child’s sex and household wealth quintiles. It implies that the associations between a child’s sex and MAMM did not vary by the household wealth quintile when all the covariates were accounted for. In other words, the household wealth status did not moderate the relationship between the child’s sex and MAMM.

Similarly, model 2 (Figure 1b), illustrated the predictive margins plot of the interaction between a child’s sex and age on MAMM among children aged 6–59 months in Nigeria. Each line in the three outcome bands represents the age groupings of the children. The non-distinct intersection of the lines indicates the non-significant presence of interaction between a child’s sex and age. However, slight variations were seen to exist for the age bands between males and females at the advantage of the female child, and these were not statistically significant. Therefore, the predictive effect of a child’s sex on MAMM did not vary by child’s age. In other words, the child’s age did not moderate the relationship between the child’s sex and MAMM. However, model 3 (model of best fit), illustrated in Figure 1c, described the results of the analysis of the two-way interaction between a child’s age and household wealth quintile in MAMM among children aged 6–59 months in Nigeria, while accounting for other covariates. Some categories in this interaction displayed statistically significant relationships with the outcome of interest. The chart shows that some of the lines’ intersections (non-parallel) indicate the existence of interaction effects. For instance, in the band of ‘none of the diseases’ (lower band) and ‘two or more diseases’ (upper band), the effects of the richest household varied at different points according to the child’s age band. So, the two-way interactions between a child’s age group and household wealth status were relevant in the model prediction of MAMM. Thus, the variation in MAMM over a child’s age differed depending on the household wealth quintile [34]. In view of the multi-categories of the predictors, the chart is non-parsimonious to describe. To help in better understanding the results from the chart, Table 3 (average adjusted probability of interaction table) was created [32,33]. For instance, in the ‘none of the diseases’ group, a child living in the richest household and aged 48–59 months had a probability of 0.31 of staying healthy from MAMM compared with a child of the same age who resides in the poorest household, with a probability of 0.12 of being healthy from MAMM. Similarly, children aged 36–47 months living in the poorest households had a probability of 0.11, 0.18, and 0.32 above the probability of children of the same age who live in the middle class, richer, and richest households, respectively, of contracting MAMM, while all other covariates were held constant.

However, model 4, which contains all covariates and all conceivable two-way interactions between a child’s sex, age, and household wealth status (i.e., the composite of models 1, 2, and 3), did not demonstrate any further improvement in the significant impacts of the interactions of child’s age and household wealth status than was observed in model 3. Furthermore, the three-way interaction between a child’s sex, age, and household wealth status considered in model 5 had no significant effects on children contracting MAMM when compared to their respective reference categories conditional on all covariates in model 4.

Table 3 further shows that the probability of children having none of the diseases for each age group is highest for children living in the wealthiest households. Similarly, the same pattern is observed in the morbidity (one disease only) group. However, the pattern changes where the probability of children of all ages with ‘two or more diseases’ decreases as the wealth quintile increases. In other words, for older age groups, the variations in MAMM become more evident between the richest and the poorest households. So, in general, the results reflect that children from the richest households in older age groups had a decreased probability of living with MAMM compared to children living in the poorest households.

## 4. Discussion

This study aimed to investigate whether a child’s sex, age, and household wealth status interact in the presence of other covariates to explain the variations in the prevalence of multimorbidity of anaemia, malaria, and malnutrition (MAMM) among children aged 6–59 months in Nigeria. An examination was conducted on how the individual and household variables at level 1, nested in the community at level 2, and nested in the state at level 3, affect the co-occurrence of multiple diseases, and how some of a child’s demographic variables (age and sex) interact with household socioeconomic status (represented by household wealth status) to modify these relationships with MAMM.

To address the aim of this study, interaction analyses investigating the relationships between the three two-way and one three-way interactions of a child’s sex, age, and household wealth status with MAMM relative to other covariates were carried out. To the best of the authors’ knowledge, this study is the foremost that has considered the relationships between the intersections of a child’s sex, age, and household wealth status and MAMM; therefore, it becomes difficult to directly compare and contrast the findings with previous studies. The results of the intersections between a child’s sex and age, a child’s sex and household wealth, and a child’s age, sex, and household wealth status were not statistically significant predictors of MAMM when adjusted for other covariates. These findings mean the following: (i) The relationship between a child’s age and MAMM did not vary by the child’s gender, and conversely, the relationship between the child’s gender and MAMM did not vary by the child’s age. The possible reason for this is that sex variations in adverse health outcomes are usually stronger than age variations. Culturally, mothers or caregivers in a typical Nigerian household often treat female and male children the same way on health ground irrespective of their age status, especially when they are above two years of age. The reasons for this could be a subject of future research. (ii) The relationship between household wealth status and MAMM did not vary by the child’s sex. What this means is that the perspectives that rich and poor households hold concerning MAMM did not differ in terms of the sex of their children. The possible reason for this is that in a typical Nigerian family, the health of the children is equally important irrespective of their gender. (iii) The inequalities in the relationship between household wealth and MAMM did not vary by the child’s age and sex. The reason for this is not known and could also be the subject of future study. However, while considering the interactions between a child’s age and household wealth status in the presence of other covariates, the results showed that the effects of household wealth status on MAMM varied by the child’s age. The implication is that children in the wealthiest households have the highest chance of maintaining a healthy status from MAMM for each age group. A similar pattern was seen in the group of children with morbidity (contracting only one disease). The likelihood of children of all age groups having ‘two or more diseases’ is lowered as the household wealth quintile rises. Compared to children living in the poorest households across all age groups, children from the richest households had the lowest chance of contracting MAMM as they became older. To put it another way, as the children become older, differences in MAMM between the rich and the poor household become more noticeable. A possible reason for this is that the richest households have the resources to care for children irrespective of their age. The older the children from the richest households, the more they are knowledgeable about what is available to eat, can ask for anything they would like to eat and receive it, can complain whenever they are unwell, and can receive medical attention promptly. The inverse is the case for children from the poorest households, where the scarce resources are used more to the advantage of younger children than older children. Older children often scavenge outside the home for survival [35], and in the process are exposed to unhealthy conditions.

The interactions studied in this paper provide strong evidence for the intersectional character of individual-level disparities in multimorbidity among children in Nigeria. Moreover, this study specifically draws attention to the partiality of results obtained from studies that only use individual characteristic models to look for patterns of individual and contextual variations in health outcomes. This position is supported by another study [26], which asserted that results of this nature lend credence to the mounting evidence of connections between social inequalities in health described in the international literature.

Moreover, a significant contribution of this study is on the premises of intersectionality investigations. A previous study highlighted that additive processes do not adequately address significant health inequalities at the point of social groupings based on the idea that variations in risk factors associated with health status are independent. Therefore, the partiality of knowledge from such a perspective raises significant questions about how well future programs can target individuals at risk of bad health outcomes [26]. Furthermore, compared to previous works, this study used an intersectionality framework to investigate the interactions of children’s demographics (age and sex) and household socioeconomic (wealth quintile) inequalities in multimorbidity while accounting for other additive covariates as a way of addressing these problems. Overall, it presents a better model fit when the interaction terms for the child’s age and household wealth quintile are considered. Also, the data set came from two nationally representative surveys with abundant evidence of hierarchy. Yet, most previous studies in these categories do not account for the multilevel structure or use the proper statistical techniques. However, this study applied multilevel methods to account for individual, community, and state variations.

However, this study is not without limitations: (i) Because the data sets were cross-sectional, they were unable to identify the distinct causes of MAMM cases in Nigerian children aged 6–59 months. This partially gives credence to the rationale of combining acute disease (malaria) and more chronic health status (anemia and malnutrition) together. A longitudinal study, which necessitates participant follow-up on a regular basis, yields more information on the causes [3]. (ii) Unfortunately, Nigeria’s high rate of maternal illiteracy, which may have caused recollection errors, so the accuracy of the information submitted at the time of the survey could not be verified. (iii) The listwise deletion approach was utilized to handle incomplete data in the variables that remained after some variables were eliminated due to missingness. For the incomplete observation, alternative techniques like multiple imputations could have been applied. Unfortunately, to the best of the authors’ knowledge, appropriate methods and coding to handle the multilevel mixed effect ordinal logistic regression effectively, and subsequently, the predictive margins plots have not been developed in STATA. (iv) The simple classifications of MAMM as counts in this study were to conform with the general definition of multimorbidity, but this has limitations in developing medical and healthcare interventions. According to the Academy of Medical Sciences, clustering of diseases would be more relevant for health intervention purposes [36]. This could be a subject of future studies.

## 5. Conclusions

Our analysis of a cross-sectional national survey of a representative sample of Nigerian households in 2018 found that overall, around 5 out of 10 children aged 6–59 months were living with two or more of the diseases of anemia, malaria, and malnutrition. This prevalence increased to 7 out of 10 children in the lowest or poorest household wealth quintile compared to 2 out of 10 children in the highest or richest household wealth quintile. This study’s findings suggest that socioeconomic conditions and household wealth should not be overlooked within the broader context of public health challenges in Nigeria. For children aged 6–59 months living among the poorest families, it is equally important that they have as much access to healthcare as their counterparts in wealthier homes. This study concludes that the richer the household and the older the children are, the less likely the children are to be living with MAMM. Policies that are geared toward socioeconomic redistribution to help the poorer households may assist in reducing the disparities observed in multiple diseases among children aged 6–59 months in Nigeria. To guide future studies and ensure the continuity of the research on MAMM in Nigeria, longitudinal studies to track changes over time, qualitative research to understand family and community dynamics, or intervention studies to test the effectiveness of specific policies will be needed. In addition, future research could include targeted interventions for at-risk populations (such as children from the poorest households), studies exploring the role of specific socioeconomic policies on health outcomes, or studies that will research the mechanisms through which a child’s age and household socioeconomic status protect against MAMM.

## Figures and Tables

**Figure 1 ijerph-21-00645-f001:**
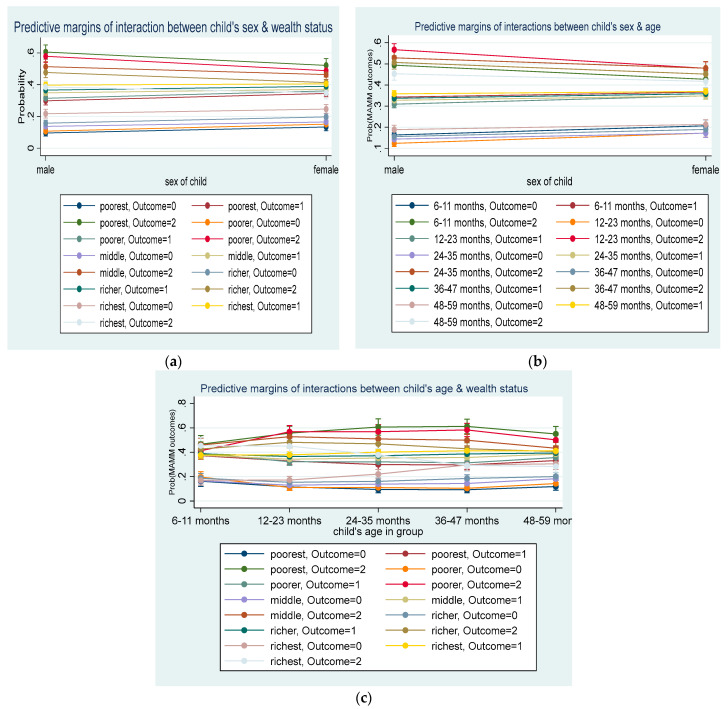
Predictive margins plot of interaction effects. (**a**) Interaction between child’s sex and household wealth quintiles; (**b**) interaction between child’s sex and age; and (**c**) interaction between child’s age and household wealth quintiles.

**Table 1 ijerph-21-00645-t001:** Characteristics and association of predictors with MAMM.

		Multimorbidity			Chi-Square
Variables	Total	None of the Diseases	One Disease Only	Two or More of the Diseases	
	*n* (%)	*n* (%)	*n* (%)	*n* (%)	
Child’s sex					χ^2^ (2) = 25.03, *p* = 0.0002
Male	5217 (51.23)	841 (16.13)	1734 (33.25)	2641 (50.62)	
Female	4967 (48.77)	926 (18.64)	1764 (35.52)	2277 (45.84)	
Child’s age in group					χ^2^ (8) = 205.55, *p* < 0.0001
6–11 months	1232 (12.1)	165 (13.35)	566 (45.91)	502 (40.74)	
12–23 months	2422 (23.78)	289 (11.95)	888 (36.66)	1245 (51.39)	
24–35 months	2160 (21.21)	363 (16.82)	694 (32.13)	1102 (51.05)	
36–47 months	2227 (21.87)	452 (20.32)	664 (29.83)	1110 (49.85)	
48–59 months	2143 (21.04)	498 (23.23)	687 (32.06)	958 (44.71)	
Household wealth index					χ^2^ (8) = 1635.53, *p* < 0.0001
Poorest	1893 (18.59)	109 (5.73)	392 (20.7)	1393 (73.57)	
Poorer	1989 (19.53)	166 (8.33)	555 (27.9)	1268 (63.77)	
Middle	2139 (21)	328 (15.35)	753 (35.19)	1058 (49.46)	
Richer	2144 (21.05)	445 (20.75)	876 (40.85)	823 (38.4)	
Richest	2019 (19.83)	720 (35.66)	924 (45.75)	376 (18.6)	
MAMM status	10,184 (100)	1767 (17.35)	3499 (34.36)	4918 (48.29)	

**Table 2 ijerph-21-00645-t002:** Distribution of model fit.

Model	Number of Covariates	Log-Likelihood	AIC	BIC
Model 1	Child’s sex * household wealth status + all the covariates (including child’s sex, age, and household wealth quintiles)	−6933.6	14,015.1	14,530.2
Model 2	Child’s sex * age + all the covariates (including child’s sex, age, and household wealth quintiles)	−6933.4	14,014.8	14,529.9
Model 3	Child’s age * household wealth status + all the covariates (including child’s sex, age, and household wealth quintiles)	−6902.6	13,977.3	14,575.9
Model 4	Child’s sex * household wealth status + child’s sex * age + child’s age * household wealth status + all the covariates (including child’s sex, age, and household wealth quintiles)	−6898.5	13,985.03	14,639.4
Model 5	Child’s sex * age * household wealth status + all the covariates (including child’s sex, age, and household wealth quintiles)	−6890.8	14,001.6	14,767.4

**Table 3 ijerph-21-00645-t003:** Average adjusted probability for interactions of child’s age and household wealth status.

Disease Classification	Age (Months)	Household Wealth Status				
		Poorest	Poorer	Middle	Richer	Richest
None of the disease	6–11	0.162	0.196	0.166	0.187	0.17
	11–23	0.116	0.111	0.129	0.153	0.172
	23–35	0.094	0.11	0.138	0.161	0.22
	35–47	0.092	0.104	0.144	0.185	0.296
	47–59	0.119	0.143	0.182	0.201	0.305
One disease only	6–11	0.372	0.392	0.375	0.387	0.377
	11–23	0.328	0.322	0.343	0.365	0.378
	23–35	0.299	0.321	0.352	0.371	0.401
	35–47	0.296	0.313	0.357	0.386	0.412
	47–59	0.331	0.356	0.385	0.394	0.412
Two or more diseases	6–11	0.466	0.412	0.459	0.426	0.454
	11–23	0.557	0.568	0.528	0.482	0.45
	23–35	0.607	0.569	0.509	0.468	0.378
	35–47	0.612	0.583	0.499	0.429	0.292
	47–59	0.55	0.501	0.433	0.406	0.283

## Data Availability

The data set used in this study is available in MeasureDHS https://dhsprogram.com (accessed on 28 January 2020) and UNDP-Nigeria http://hdr.undp.org/sites/default/files/hdr_2018_nigeria_finalfinalx3.pdf (accessed on 3 March 2020).

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
