# Peer review of "The Intersection of a Child’s Demographics and Household Socioeconomic Status in the Multimorbidity of Malaria, Anaemia, and Malnutrition among Children Aged 6–59 Months in Nigeria"

_ijerph, 2024, doi:10.3390/ijerph21050645_

Round 1

Reviewer 1 Report

Comments and Suggestions for Authors

Abstract

This should be clear on the socioeconomic status. This should be the household socioeconomic status.

Introduction

This section is not clear. It lacks coherence and does not introduce the subject matter properly.

Methods and Data

The covariates controlled for in this study should be listed and defined.

Why the use of a second dataset when the study variables can be found in one of them?

The DHS recode used for this study should be specified.

Discussion

The first paragraph refers to child’s socioeconomic status, this should be corrected.

Last statement 2nd paragraph “The older children often wander around to scavenge outside the home for survival [35], and in the process are exposed to unhealthy conditions.”  I think the choice of words for this statement is a bit harsh though supported by a finding. It doesn’t sit well to say an under-five child who belongs to a household (based on the survey) will be scavenging for survival. you may want to check again.

Overall, it is a nice work that addresses an important public health issue. However, the manuscript can be improved.

Comments on the Quality of English Language

You may also need the service of a language editor.

Author Response

Please find in the attached file the responses to your comments. Thank you

Reviewer 2 Report

Comments and Suggestions for Authors

This study aims to determine whether child’s sex, age and socioeconomic status in-teract in the presence of other covariates to explain the variations in Multimorbidity of malaria, anaemia, and malnutrition (MAMM) among children age 6-59 months in Nigeria. a significant contribution of this study is on the premises of intersection-ality investigation made.  Furthermore compared to previous works used an intersectionality framework to investigate the interactions of children's demographics (age and sex) and household socioeconomic (wealth quintiles) inequalities in multimorbidity while ac-counting for other additive covariates as a way in addressing these problems.Unfortunately Nigeria's high rate of maternal illiteracy, this research it which may have caused recollection errors submitted at the time of the survey could not be verified.

Author Response

Please see the attached file for the responses to reviewer's comments
